# Low Levels of Low-Density Lipoprotein Cholesterol and Endothelial Function in Subjects without Lipid-Lowering Therapy

**DOI:** 10.3390/jcm9123796

**Published:** 2020-11-24

**Authors:** Yuji Takaeko, Masato Kajikawa, Shinji Kishimoto, Takayuki Yamaji, Takahiro Harada, Yiming Han, Yasuki Kihara, Eisuke Hida, Kazuaki Chayama, Chikara Goto, Yoshiki Aibara, Farina Mohamad Yusoff, Tatsuya Maruhashi, Ayumu Nakashima, Yukihito Higashi

**Affiliations:** 1Department of Cardiovascular Medicine, Graduate School of Biomedical and Health Sciences, Faculty of Medicine, Hiroshima University, Hiroshima 734-8553, Japan; y.takaeko@gmail.com (Y.T.); shinji0922k@yahoo.co.jp (S.K.); ts5216yt@gmail.com (T.Y.); harataka0513@gmail.com (T.H.); tuplev144@gmail.com (Y.H.); ykihara@hiroshima-u.ac.jp (Y.K.); 2Division of Regeneration and Medicine, Medical Center for Translational and Clinical Research, Hiroshima University Hospital, Hiroshima 734-8551, Japan; m-kajikawa@hiroshima-u.ac.jp; 3Department of Biostatistics and Data Science, Graduate School of Medicine, Faculty of Medicine, Osaka University, Osaka 565-0871, Japan; e-hida@bsds.med.osaka-u.ac.jp; 4Department of Gastroenterology and Metabolism, Graduate School of Biomedical and Health Sciences, Hiroshima University, Hiroshima 734-8551, Japan; chayama@mba.ocn.ne.jp; 5Department of Physical Therapy, Hiroshima International University, Hiroshima 739-2695, Japan; t-goto@hs.hirokoku-u.ac.jp; 6Department of Cardiovascular Regeneration and Medicine, Research Institute for Radiation Biology and Medicine, Hiroshima University, Hiroshima 734-8553, Japan; fairytale2007001@yahoo.co.jp (Y.A.); drfarinamyusoff@gmail.com (F.M.Y.); 55maruchin@gmail.com (T.M.); 7Department of Stem Cell Biology and Medicine, Graduate School of Biomedical and Health Sciences, Faculty of Medicine, Hiroshima University, Hiroshima 734-8553, Japan; ayumu@hiroshima-u.ac.jp

**Keywords:** low-density cholesterol, endothelial function, flow-mediated vasodilation, Hypolipidemia, atherosclerosis

## Abstract

An elevation of serum low-density lipoprotein cholesterol (LDL-C) levels has been associated with endothelial dysfunction in statin naïve subjects. However, there is no information on endothelial function in subjects with extremely low levels of LDL-C. The purpose of the present study was to determine the relationship of LDL-C levels, especially low levels of LDL-C, with endothelial function. Endothelial function assessed by flow-mediated vasodilation (FMD) measurement and LDL-C levels were evaluated in 7120 subjects without lipid-lowering therapy. We divided the subjects into five groups by LDL-C levels: <70 mg/dL, 70–99 mg/dL, 100–119 md/dL, 120–139 mg/dL, and ≥140 mg/dL. FMD values were significantly smaller in subjects with LDL-C levels of ≥140 mg/dL than in those with LDL-C levels of 70–99 mg/dL and 100–119 mg/dL (*p* < 0.001 and *p* = 0.004, respectively). The FMD values in the LDL-C of <70 mg/dL group were not significantly different from those in the other groups. To evaluate the relationship of extremely low LDL-C levels with endothelial function, we divided the subjects with LDL-C of <70 mg/dL into those with LDL-C levels of <50 mg/dL and 50–69 mg/dL. FMD values were similar in the LDL-C <50 mg/dL group and ≥50 mg/dL group in the propensity score-matched population (*p* = 0.570). A significant benefit was not found in subjects with low LDL-C levels from the aspect of endothelial function.

## 1. Introduction

Atherosclerotic cardiovascular disease (ASCVD) is the cause of morbidity and mortality worldwide [1]. An elevation of serum low-density lipoprotein cholesterol (LDL-C) levels is a risk factor of ASCVD [2]. It has been shown in large prospective epidemiologic cohort studies that there is a positive relationship between plasma LDL-C concentrations and ASCVD risk [3,4]. Randomized controlled trials have shown that lowering elevated LDL-C using statins significantly reduces the risk of major cardiovascular events [5,6,7,8]. These findings from epidemiologic and clinical studies indicate that elevation of LDL-C causes ASCVD [9]. It has also been shown that statins have pleiotropic effect. Statin therapy is recommended for cardiovascular disease in the 2018 American College Cardiology (ACC)/America Heart Association (AHA) cholesterol guidelines [10,11]. Higher dosages of statins for therapy are recommended for secondary prevention, and the effectiveness of drugs that lower LDL-C more strongly, including proprotein convertase subtilisin/kexin type 9 (*PCSK9*) inhibitor and ezetimibe, has been shown [12,13,14]. With the emergence of these new powerful LDL-lowering drugs, the excessive decrease in LDL-C levels is expected.

However, several studies have shown that high LDL-C levels are a good marker of longevity rather than a risk factor of mortality [15,16,17]. Moreover, a recent Korean epidemiological study has shown that low LDL-C levels (<70 mg/dL) without lipid-lowering agents were independently associated with the increase in risk of mortality and even cardiovascular disease mortality [18]. In a prospective observational study, subjects with low LDL-C levels (<70 mg/dL) had a significantly higher risk of intracerebral hemorrhage than did those with LDL-C levels of 70–99 mg/dL, and subjects with LDL-C levels of <50 mg/dL had the highest hazard ratio for risk of intracerebral hemorrhage [19]. Most randomized trials have been for subjects with LDL-C above a certain level, and subjects with very low LDL-C level have been excluded from analysis [12,13,14,20]. The degrees of atherosclerosis and cardiovascular mortality of subjects with extremely low LDL-C levels without lipid-lowering agents have been not elucidated.

The endothelium plays a very important role in maintenance of vascular homeostasis [21]. Endothelial cells synthesize various physiologically active substances including nitric oxide (NO) and these substances cause vascular smooth muscle relaxation and vasodilation [22]. Measurement of flow-mediated vasodilation (FMD) has been used as an assessment of endothelial function [23]. Endothelial dysfunction would be the initial step of atherosclerosis, leading to cardiovascular disease [24,25]. In addition, endothelial dysfunction assessed by impaired FMD is associated with future cardiovascular events [26,27].

We previously showed that vascular function was impaired in statin-naïve subjects with LDL-C levels of >100 mg/dL [28]. However, a significant benefit was not observed in subjects with LDL-C levels of <70 mg/dL. In addition, there is no information on endothelial function in subjects with extremely low levels of LDL-C (<50 mg/dL). Therefore, we evaluated the relationship of low levels of LDL-C with endothelial function in subjects without lipid-lowering therapy.

## 2. Materials and Methods

### 2.1. Subjects

A total of 10,247 subjects were enrolled from the FMD Japan registry (*n* = 7385) and from the Hiroshima University Vascular Function registry (*n* = 2862) between August 2007 and August 2016. The FMD-J study was a prospective multicenter study conducted at 22 university hospitals and affiliated clinics in Japan to examine the usefulness of FMD in risk stratification for cardiovascular disease in Japanese subjects. The rationale and design of the FMD-J study have been described previously [29]. Subjects with a history of cardiovascular disease (*n* = 1326), subjects without information on LDL-C levels (*n* = 728), subjects with triglycerides of more than 400 mg/dL (*n* = 40), subjects in whom FMD had not been measured accurately (*n* = 1), subjects who were taking lipid-lowering medicine (*n* = 1101), and subjects with an estimated glomerular filtration rate (eGFR) < 15 mL/min/1.73 m^2^ (*n* = 6) were excluded from this study. Finally, 7120 subjects were enrolled in this study (Appendix A). Cardiovascular disease was defined as cerebrovascular disease and coronary heart disease. Cardiovascular disease including included transient ischemic attack, ischemic stroke, and hemorrhagic stroke was defined as cerebrovascular disease and coronary heart disease. Coronary heart disease included angina pectoris, unstable angina, and myocardial infarction. Hypertension was defined as systolic blood pressure of 140 mm Hg or higher, diastolic pressure of 90 mm Hg or higher, or treatment with oral antihypertensive drugs. Diabetes mellitus was defined by the American Diabetes Association recommendation [30]. Dyslipidemia was defined by the third report of the National Cholesterol Education Program [31]. The Framingham risk score was calculated on the basis of the risk factors, including age, gender, systolic blood pressure, total cholesterol level, high-density lipoprotein cholesterol (HDL-C) level, diabetes mellitus, and smoking habit [32]. LDL-C concentration was calculated using the Friedewald equation. The protocol of the current study was approved by the ethical committees of the participating institutions and registered in the University Hospital Medical Information Network Clinical Trials Registry (UMIN000012950). Informed consent was obtained from all of the participants.

### 2.2. Measurement of FMD

Subjects fasted overnight for at least 12 h and abstained from caffeine, alcohol, smoking, and antioxidant vitamins on the day of the FMD examination [27,28]. The subjects were kept in the supine position in a quiet, dark, air-conditioned room (constant temperature of 22–25 °C) through the study [27,28]. The protocol of FMD measurement was previously described [27,28]. Briefly, vascular response to reactive hyperemia in the brachial artery was used for assessment of endothelium-dependent FMD. We used a high-resolution ultrasonography system (UNEXEF18G, UNEX Co, Nagoya, Japan) for evaluation of FMD. The brachial artery was scanned longitudinally 5 to 10 cm above the elbow. When the clearest B-mode image of the anterior and posterior intimal interfaces between the lumen and vessel wall was obtained, the transducer was held at the same point throughout the scan by a special probe holder (UNEX Co) to ensure consistency of the image. When the tracking gate was placed on the intima, the artery diameter was automatically tracked, and the waveform of diameter changes over the cardiac cycle was displayed in real-time using the FMD mode of the tracking system [27,28]. This allowed the ultrasound images to be optimized at the start of the scan and the transducer position to be adjusted immediately for optimal tracking performance throughout the scan. Baseline longitudinal imaging of the artery was performed for 30 s, and then the blood pressure cuff placed around the forearm was inflated to 50 mmHg above systolic pressure for 5 min. Longitudinal imaging of the artery was recorded continuously until 5 min after cuff deflation. At the same time as the recording of the longitudinal image of the artery, we also observed the short axis image using a high resolution ultrasonography system. Changes in brachial artery diameter were immediately expressed as the percentage change relative to the vessel diameter before cuff inflation [27,28]. FMD was automatically calculated as the percentage change in the peak vessel diameter from the baseline value. The percentage of FMD (peak diameter—baseline diameter)/baseline diameter) was used for analysis. The correlation coefficient between FMD analyzed at the core laboratory and at participant institutions was 0.84 (*p* < 0.001) [27,28]. We confirmed that the Bland–Altman analysis revealed no systemic bias in the variability of FMD measurement between the participating institutions and the core laboratory [33]. The observers were blind to the form of examination.

### 2.3. Statistical Analysis

Results are presented as means ± SDs or medians (interquartile range) for continuous variables and as percentages for categorical variables. All reported probability values were two-sided. A probability value of <0.05 was considered statistically significant. In accordance with the LDL-C quartiles and current guidelines, subjects were divided into five groups by LDL-C levels. To assess the association of extremely low LDL-C levels with endothelial function, the subjects with an LDL-C of <70 mg/dL were divided into <50 mg/dL and 50–69 mg/dL groups according to another prospective study [19]. Comparison of variables among two or more groups by differences in the LDL-C levels was performed using one-way analysis of variance (ANOVA). Categorical values, such as medications and medical histories, were compared by means of the χ^2^ test. Tukey’s post hoc test was used to compare the differences in FMD between these groups. As a sensitivity analysis, propensity score analysis was used to generate a set of matched cases and all remaining participants of this study. A logistic regression model was used to estimate the propensity of LDL-C levels < 70 mg/dL based on variables associated with LDL-C, including age, body mass index, gender, heart rate, glucose, triglycerides, HDL-C, hypertension (yes or no), diabetes mellitus (yes or no), smokers (yes or no), use of anti-hypertensive drugs (yes or no), and use of anti-hyperglycemic therapy (yes or no). With these propensity scores, using a caliper width of 0.2 SDs of the logit of the propensity score, two well-matched groups based on clinical characteristics were created for comparison of the ratio of FMD < 4.0%, which was the division point for the lowest FMD quartile in all subjects. All of the statistical analyses were conducted using JMP version 13.0 software (SAS Institute, Cary, NC, USA) and Stata version 17 software (Stata Corporation, College Station, TX, USA).

## 3. Results

### 3.1. Baseline Characteristics

The quartiles of LDL-C were 99 mg/dL, 118 mg/dL (median), and 139 mg/dL. According to these quartiles and current guidelines, we divided the subjects into five groups by LDL-C levels: < 70 mg/dL, 70–99 mg/dL, 100–119 mg/dL, 120–139 mg/dL, and ≥ 140 mg/dL. Baseline characteristics of all of the subjects according to LDL-C levels are summarized in Table 1. Of the 7120 subjects, 5465 (76.8%) were men, 2924 (41.1%) had hypertension, 2986 (41.9%) had dyslipidemia, 511 (7.2%) had diabetes mellitus, and 2165 (30.5%) were current smokers. There were significant differences among the five groups in age, body mass index, systolic blood pressure, diastolic blood pressure, heart rate, total cholesterol, triglycerides, HDL-C, glucose, use of anti-hypertensive drugs, prevalence of hypertension and dyslipidemia, and percentage of smokers.

### 3.2. Relationship of LDL-C with Endothelial Function

There was a significant inverse relationship between FMD and LDL-C levels (*r* = −0.07, *p* < 0.001) (Figure 1 and Appendix A). Multiple regression analyses revealed that LDL-C was an independent predictor of FMD (Table 2). FMD values were significantly smaller in the LDL-C ≥ 140 mg/dL group than in the 70–99 mg/dL group and the 100–119 mg/dL group (*p* < 0.001 and *p* = 0.004, respectively; Figure 2). FMD values were also significantly smaller in the 120–139 mg/dL group than in the 70–99 mg/dL group and the 100–119 mg/dL group (*p* < 0.001 and *p* = 0.008, respectively; Figure 2). Figure 1 shows the relationship between FMD and LDL-C.

### 3.3. Relationship of Extremely Low LDL-C with Endothelial Function

The FMD values in the LDL-C < 70 mg/dL group were not significantly different from those in the other groups (Table 1 and Figure 2). As a sensitivity analysis, propensity score analysis was used to generate a set of matched cases (subjects with LDL-C levels of <70 mg/dL) and all remaining participants of this study (≥70 mg/dL). A logistic regression model was used to estimate the propensity of LDL-C levels < 70 mg/dL based on variables associated with LDL-C, including age, body mass index, gender, heart rate, glucose, triglycerides, HDL-C, hypertension (yes or no), diabetes mellitus (yes or no), smokers (yes or no), use of anti-hypertensive drugs (yes or no), and use of anti-hyperglycemic therapy (yes or no). With these propensity scores, using a caliper width of 0.2 standard deviations of the logit of the propensity score, two well-matched groups based on clinical characteristics were created for comparison of the ratio of FMD < 4.0%, which was the division point for the lowest quartile of FMD in all subjects. FMD was not significantly different between the LDL-C < 70 mg/dL group and ≥70 mg/dL group in the propensity score-matched population (*p* = 0.386) (Table 3). To evaluate the relationship of extremely low LDL-C levels with endothelial function, we divided the subjects with an LDL-C of <70 mg/dL into a <50 mg/dL group (*n* = 35) and a 50–69 mg/dL group (*n* = 210). Baseline characteristics are summarized in Table 4. There were significant differences among the six groups in age, body mass index, systolic blood pressure, diastolic blood pressure, heart rate, total cholesterol, triglycerides, HDL-C, glucose, use of anti-hypertensive drugs, prevalence of hypertension and dyslipidemia, and percentage of smokers.

There were no significant differences among the LDL-C < 50 mg/dL group, LDL-C 50–69 mg/dL group, and LDL-C 70–99 mg/dL group in FMD, and there were no significant differences among the LDL-C < 50 mg/dL group, LDL-C 50–69 mg/dL group, and LDL-C ≥ 140 mg/dL group in FMD (Appendix A). The propensity score analysis was used to compare the <50 mg/dL group with all remaining participants. The variables used for the propensity score were the same as those described earlier. There was no significant difference in FMD values between the LDL-C < 50 mg/dL group and ≥ 50 mg/dL group in the propensity score-matched population (*p* = 0.570) (Appendix A).

## 4. Discussion

In the present study, there was a significant inverse relationship between FMD and LDL-C levels in subjects without cardiovascular disease and not receiving lipid-lowering therapy. Although there was not much difference in FMD among the groups categorized by LDL-C levels, FMD was highest in subjects with LDL-C of 70–99 mg/dL. FMD in the low LDL-C groups, including <70 mg/dL and <50 mg/dL groups, were not significantly different from that in the other groups. FMD values were similar in the low LDL-C groups and all remaining participants in the propensity score-matched population.

In several epidemiological studies, there was an inverse relationship between total cholesterol and all-cause mortality, which could not be explained by reverse causality (i.e., serious diseases cause low cholesterol) [15,16,17]. A recent epidemiological study has shown the association of low levels of LDL-C (<70 mg/dL) with the increase in risk of cardiovascular disease in healthy Korean subjects without lipid-lowering therapy [18]. In high-risk primary prevention individuals in the REGARDS study, low high-sensitivity C reactive protein levels of <2 mg/dL appeared to be related to the decrease in risk of cardiovascular complications, whereas low LDL-C levels of <70 mg/dL were not related to prevention of cardiovascular complications [34]. Results of a 9-year follow-up of 96043 participants without cardiovascular complications and cancer at baseline showed that subjects with LDL-C levels <70 mg/dL had a higher risk of intracerebral hemorrhage than did those with LDL-C levels of 70–99 mg/dL, and adjusted hazard ratios were 1.65 for LDL-C levels of 50–69 mg/dL and 2.69 for LDL-C levels <50 mg/dL [19]. It is likely that low LDL-C levels do not always mean longevity and low cardiovascular morbidity.

In statin-naïve subjects, several investigators have reported an association of LDL-C levels with endothelial function assessed by FMD [28,35]. In a cross-sectional observational study of relatively young and healthy military men, FMD was correlated linearly with LDL-C, and FMD values were highest in subjects with an LDL-C of <100 mg/dL [35]. Our previous study showed that vascular function is impaired in subjects with an LDL-C of >100 mg/dL and that the optimal cutoff level for maintenance of vascular function is an LDL-C of 100 mg/dL [28]. In the present study, FMD values were highest in subjects with LDL levels of 70–99 mg/dL, consistent with the results of previous studies [28,35].

There are many reports showing that statin treatment improved endothelial function [36,37,38,39]. As the underlying mechanism, it has been shown that long-term statin treatment improves endothelial function via inhibition of vascular cell senescence, amelioration of oxidative stress, and normalization of endothelial and inducible NO synthase imbalance [40]. Statins have not only LDL-C-lowering effects but also pleiotropic effects for atherosclerosis [10]. A meta-analysis revealed that most of the anti-inflammatory effects of statins were related to the magnitude of LDL reduction [41]. On the other hand, there are some reports showing that endothelial function was not improved by treatment with a statin [42,43,44]. In patients with diabetes mellitus without manifest cardiovascular disease, 2-year statin therapy had no effect on FMD [42]. In patients with diabetes mellitus and dyslipidemia without cardiovascular disease, intensive lipid-lowering therapy by atorvastatin did not have beneficial effects on endothelial function [43]. In patients with insulin resistant familial combined hyperlipidemia, 12-week treatment with rosuvastatin did not improve endothelial function despite plasma cholesterol being significantly decreased [44]. In the present study, we selected subjects with reduced LDL-C who were not taking lipid-improving agents including statins. We were not able to directly compare our results with the results of studies using statins. In a previous study, there was no significant association of LDL-C levels with FMD in subjects receiving statins.

Meta-analyses of data from 27 randomized clinical trials in the Cholesterol Treatment Trialists’ Collaboration showed that statins decreased the risk of cardiovascular events by 21% for each 38.7 mg/dL reduction in LDL-C [45]. The efficacy of statin use for prevention of cardiovascular events has been established. We do not deny the beneficial effects of lowering LDL-C to <50 mg/dL by statin treatment on cardiovascular events. Indeed, the Improved Reduction of Outcomes: Vytorin Efficacy International Trial (IMPROVE-IT) showed that maintenance of a mean LDL-C level of 53.7 mg/dL by treatment with a statin and ezetimibe improved cardiovascular events compared to a mean LDL-C level of 69.5 mg/dL [12]. The further Cardiovascular Outcomes Research with PCSK9 Inhibition in Subjects with Elevated Risk (FOURIER) trial showed that a decrease in LDL-C level from 92 to 30 mg/dL by treatment with a PCSK-9 inhibitor in combination with statins decreased cardiovascular events by about 15% [13]. These findings suggest that for the prevention of cardiovascular events, the rule for the decrease in LDL-C levels is the earlier the better and the lower the better. We may have to consider separately the effects of LDL-C levels with lipid-lowering therapy and natural LDL-C levels on endothelial function and cardiovascular events.

There are a number of limitations in the present study. First, this study was a cross-sectional study. We could evaluate the association of LDL-C levels with endothelial function but not causality. Second, we did not know the accurate causes of extremely low LDL-C in the present study. In order to evaluate subjects without secondary causes of low LDL-C, we excluded subjects who were taking lipid-lowering medicine and eGFR < 15 mL/min/1.73 m^2^, BMI < 16 kg/m^2^, malignant tumor, and active infection. However, there may be residual unrecognized confounders between decrease in LDL-C and endothelial dysfunction. Confounding factors may remain even after the propensity score match. Our database had limited information on physical activity and educational status. These confounders can affect the relationship between FMD and LDL-C levels. In addition, lipoprotein disorders in which LDL-C concentrations are genetically reduced have been reported, and among them, patients with mutations in the *ANGPTL3* and *PCSK9* genes are asymptomatic [46]. Although sequence variants in the *PCSK9* gene were associated with reduced LDL-C levels and reduced risk of cardiovascular events, LDL-C levels were approximately 100 mg/dL, which is not an extremely low level such as a level of <70 mg/dL or <50mg/dL [47]. One case of a homozygote for *PCSK9* nonsense mutations that had an extremely low LDL-C concentration of approximately 16 mg/dL was reported [48]. However, there has been no report on the relationship between *PCSK9* nonsense mutations and early atherosclerosis. Mutations of loss-of-function in the *ANGPTL3* gene are associated with hypolipidemia, which is characterized by low levels of HDL-C, LDL-C, and triglycerides [49]. Although heterozygous *ANGPTL3* loss-of-function variants were associated with a 41% lower odds of coronary artery disease, the median level of LDL-C in the heterozygous variants was approximately 112 mg/dL [50]. It has been reported that homozygotes for *ANGPTL3* have been associated with the increase in carotid intima-media thickness and decreased FMD despite low plasma LDL-C levels (approximately 63 mg/dL) [51]. This increased risk for the development of early vascular atherosclerotic changes might be related to impaired HDL function. Assessment of genetic abnormalities related to LDL-C levels would draw more specific conclusions regarding the role of low LDL-C in vascular function. Third, in the present study, there was only a small number of subjects with extremely low LDL-C levels (two subjects with LDL-C levels of <30 mg/dL). Further studies involving a large number of subjects with extremely low LDL-C levels are needed to evaluate the relationship between low LDL-C levels and endothelial function.

## 5. Conclusions

In conclusion, FMD values were highest in subjects with LDL-C of 70–99 mg/dL. In the propensity score-matched population, FMD values were similar in the low LDL-C groups and all remaining participants. Although the precise mechanisms of the relationship between low LDL-C and endothelial function is not known, a significant benefit was not found in subjects with low LDL-C levels who were not receiving lipid-lowering treatment and had no history of cardiovascular disease from the aspect of endothelial function.

## Figures and Tables

**Figure 1 jcm-09-03796-f001:**
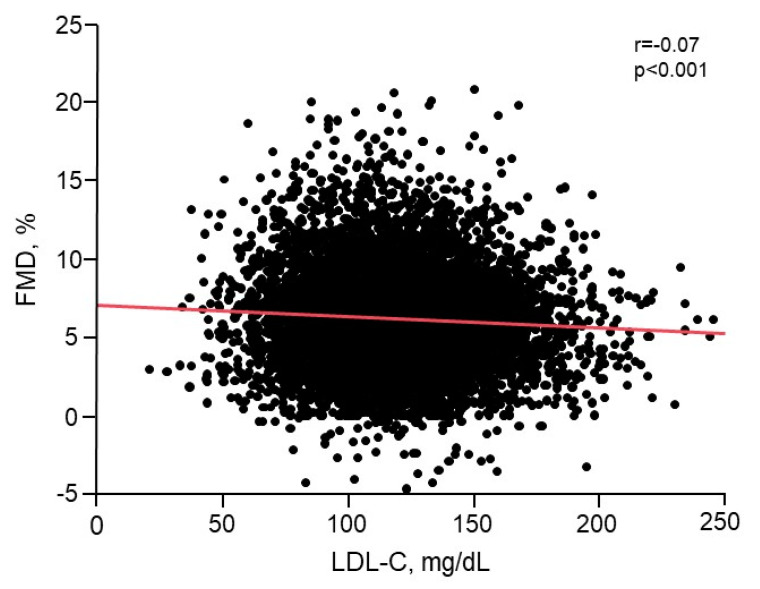
Scatter plot showing the relationship between FMD and LDL-C.

**Figure 2 jcm-09-03796-f002:**
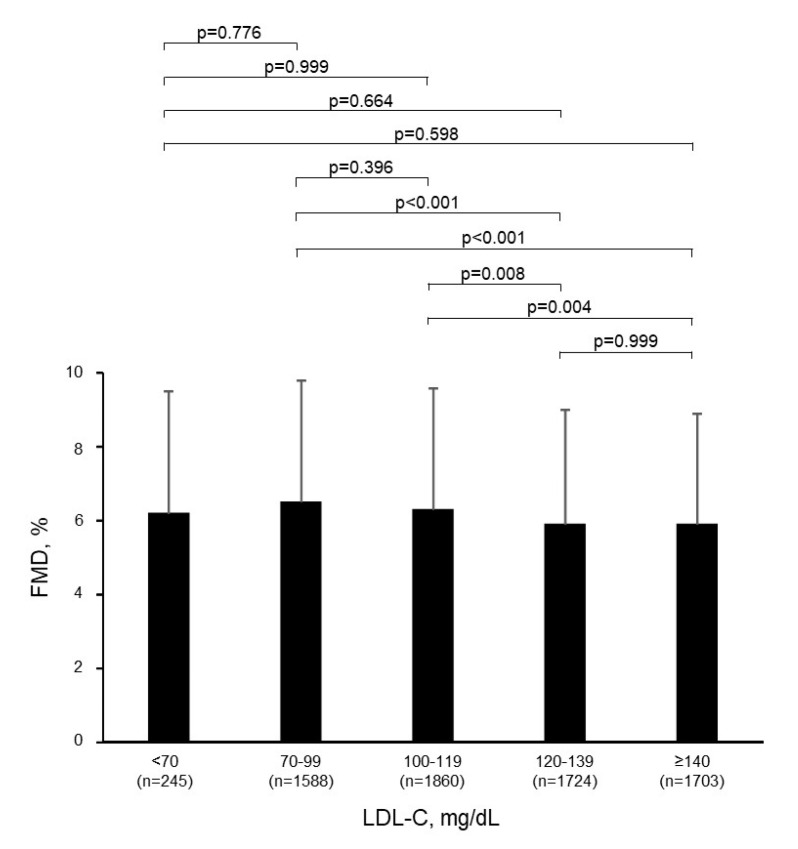
Comparison of FMD values of each group. Bar graphs show flow-mediated vasodilation (FMD) in low-density lipoprotein cholesterol (LDL-C) < 70 mg/dL, 70−99 mg/dL, 100−119 mg/dL, 120−139 mgdL, and ≥ 140 mg/dL groups. The error bars indicate the standard deviation.

**Table 1 jcm-09-03796-t001:** Clinical characteristics of the subjects on the basis of LDL-C.

Variables	Total (*n* = 7120)	<70 mg/dL (*n* = 245)	70–99 mg/dL (*n* = 1588)	100–119 mg/dL (*n* = 1860)	120–139 mg/dL (*n* = 1724)	≥140 mg/dL (*n* = 1703)	*p*-Value
Age, year	50.2 ± 12	47.5 ± 14	47.1 ± 13	50.4 ± 12	51.2 ± 11	52.3 ± 10	<0.001
Body mass index, kg/m^2^	23.3 ± 3.4	22.2 ± 3.4	22.3 ± 3.3	23.1 ± 3.3	23.8 ± 3.3	24.1 ± 3.4	<0.001
Gender, men/women	5465/1655	182/63	1222/366	1428/432	1361/363	1272/431	0.050
Systolic blood pressure, mmHg	127 ± 17	125 ± 19	124 ± 17	127 ± 17	129 ± 16	130 ± 17	<0.001
Diastolic blood pressure, mmHg	79 ± 12	78 ± 13	77 ± 12	79 ± 12	81 ± 12	81 ± 12	<0.001
Heart rate, bpm	65 ± 11	66 ± 13	64 ± 11	65 ± 11	64 ± 10	66 ± 11	<0.001
Total cholesterol, mg/dL	203 ± 33	146 ± 23	172 ± 19	193 ± 16	211 ± 16	242 ± 23	<0.001
Triglycerides, mg/dL	104 (72, 149)	87 (57, 145)	85 (61, 130)	95 (67, 136)	111 (79, 155)	123 (90, 162)	<0.001
HDL-C, mg/dL	60 ± 16	64 ± 20	63 ± 17	61 ± 15	58 ± 15	57 ± 14	<0.001
LDL-C, mg/dL	120 ± 30	60 ± 9	88 ± 8	110 ± 6	129 ± 6	159 ± 18	-
Glucose, mg/dL	100 ± 20	100 ± 21	97 ± 18	100 ± 21	100 ± 19	102 ± 22	<0.001
Medications, *n* (%)							
Anti-hypertensive therapy	1669 (23.5)	59 (24.0)	312 (19.7)	443 (23.8)	460 (26.7)	395 (23.2)	<0.001
Anti-hyperglycemic therapy	311 (4.4)	16 (6.5)	70 (4.4)	80 (4.3)	70 (4.1)	75 (4.4)	0.544
Framingham risk score, %	8.1 ± 7.2	2.7 ± 2.2	3.3 ± 2.9	7.9 ± 6.4	9.4 ± 7.0	12.3 ± 8.5	<0.001
Medical history, *n* (%)							
Hypertension	2924 (41.1)	90 (36.6)	543 (34.2)	767 (41.2)	769 (44.6)	755 (44.3)	<0.001
Dyslipidemia	2986 (41.9)	63 (25.7)	320 (20.2)	392 (21.1)	508 (29.5)	1703 (100.0)	<0.001
Diabetes mellitus	511 (7.2)	22 (9.0)	96 (6.1)	132 (7.1)	117 (6.8)	144 (8.5)	0.066
Smokers	2165 (30.5)	82 (33.5)	530 (33.4)	542 (29.2)	524 (30.5)	487 (28.7)	0.021
FMD, %	6.1 ± 3.2	6.2 ± 3.3	6.5 ± 3.3	6.3 ± 3.3	5.9 ± 3.1	5.9 ± 3.0	<0.001

HDL-C indicates high-density lipoprotein cholesterol; LDL-C, low-density lipoprotein cholesterol; FMD, flow-mediated vasodilation.

**Table 2 jcm-09-03796-t002:** Multiple linear regression analysis of the relationships between FMD and variables.

Variables	β	VIF	Parameter Estimate	Standard Error	*p*-Value
Age, year	−0.284	1.45	−0.077	0.004	<0.001
Body mass index, kg/m^2^	−0.114	1.30	−0.109	0.012	<0.001
Systolic blood pressure, mmHg	−0.112	2.85	−0.021	0.004	<0.001
Diastolic blood pressure, mmHg	0.074	2.66	0.020	0.005	<0.001
Heart rate, bpm	0.048	1.11	0.014	0.003	<0.001
Total cholesterol, mg/dL	0.860	70.45	0.083	0.009	<0.001
Triglycerides, mg/dL	−0.386	10.69	−0.019	0.002	<0.001
HDL-C, mg/dL	−0.420	16.78	−0.085	0.009	<0.001
LDL-C, mg/dL	−0.744	58.13	−0.080	0.009	<0.001
Glucose, mg/dL	−0.035	1.21	−0.006	0.002	0.004
Framingham risk score, %	−0.036	2.40	−0.016	0.008	0.037

The adjusted r^2^ was 0.16. FMD indicates flow-mediated vasodilation; HDL-C, high-density lipoprotein cholesterol; LDL-C, low-density lipoprotein cholesterol; VIF, variance inflation factor.

**Table 3 jcm-09-03796-t003:** Clinical characteristics in propensity score-matched subjects.

Variables	<70 mg/dL (*n* = 244)	≥70 mg/dL (*n* = 244)	*p*-Value
Age, year	47.5 ± 14	47.5 ± 13	0.997
Body mass index, kg/m^2^	22.2 ± 3.4	22.2 ± 3.5	0.856
Gender, men/women	182/61	196/47	0.127
Systolic blood pressure, mmHg	125 ± 19	126 ± 17	0.354
Diastolic blood pressure, mmHg	78 ± 13	78 ± 12	0.795
Heart rate, bpm	66 ± 13	66 ± 11	0.619
Total cholesterol, mg/dL	147 ± 22	203 ± 32	<0.001
Triglycerides, mg/dL	87 (56, 145)	95 (67, 144)	0.826
HDL-C, mg/dL	65 ± 20	65 ± 19	0.998
LDL-C, mg/dL	60 ± 9	117 ± 28	<0.001
Glucose, mg/dL	100 ± 22	99 ± 23	0.759
Medications, *n* (%)			
Anti-hypertensive therapy	59 (24.3)	59 (24.3)	1.000
Anti-hyperglycemic therapy	16 (6.6)	17 (7.0)	0.857
Framingham risk score, %	2.7 ± 2.1	7.2 ± 6.9	<0.001
Medical history, *n* (%)			
Hypertension	90 (37.0)	82 (33.7)	0.448
Dyslipidemia	62 (25.5)	84 (34.6)	0.030
Diabetes mellitus	22 (9.1)	24 (9.9)	0.757
Smokers	82 (33.7)	79 (32.5)	0.773
FMD, %	6.2 ± 3.2	6.4 ± 3.3	0.386

Data are presented as mean ± SD or median (interquartile range). HDL-C indicates high-density lipoprotein cholesterol; LDL-C, low-density lipoprotein cholesterol; FMD, flow-mediated vasodilation. Variables used for propensity score-matched analysis: age, body mass index, gender, heart rate, glucose, triglycerides, HDL-C, hypertension (yes or no), diabetes mellitus (yes or no), smokers (yes or no), use of anti-hypertensive drugs (yes or no), and use of anti-hyperglycemic therapy (yes or no).

**Table 4 jcm-09-03796-t004:** Clinical characteristics of the subjects on the basis of LDL-C.

Variables	Total (*n* = 7120)	<50 mg/dL (*n* = 35)	50–69 mg/dL (*n* = 210)	70–99 mg/dL (*n* = 1588)	100–119 mg/dL (*n* = 1860)	120–139 mg/dL (*n* = 1724)	≥140 mg/dL (*n* = 1703)	*p*-Value
Age, year	50.2 ± 12	49.1 ± 13	47.2 ± 14	47.1 ± 13	50.4 ± 12	51.2 ± 11	52.3 ± 10	<0.001
Body mass index, kg/m^2^	23.3 ± 3.4	22.5 ± 4.0	22.1 ± 3.3	22.3 ± 3.3	23.1 ± 3.3	23.8 ± 3.3	24.1 ± 3.4	<0.001
Gender, men/women	5465/1655	28/7	154/56	1222/366	1428/432	1361/363	1272/431	0.067
Systolic blood pressure, mmHg	127 ± 17	126 ± 21	124 ± 18	124 ± 17	127 ± 17	129 ± 16	130 ± 17	<0.001
Diastolic blood pressure, mmHg	79 ± 12	78 ± 13	78 ± 13	77 ± 12	79 ± 12	81 ± 12	81 ± 12	<0.001
Heart rate, bpm	65 ± 11	66 ± 11	66 ± 13	64 ± 11	65 ± 11	64 ± 10	66 ± 11	<0.001
Total cholesterol, mg/dL	203 ± 33	132 ± 22	149 ± 22	172 ± 19	193 ± 16	211 ± 16	242 ± 23	<0.001
Triglycerides, mg/dL	104 (72, 149)	87 (65, 168)	87 (56, 141)	85 (61, 130)	95 (67, 136)	111 (79, 155)	123 (90, 162)	<0.001
HDL-C, mg/dL	60 ± 16	65 ± 21	64 ± 20	63 ± 17	61 ± 15	58 ± 15	57 ± 14	<0.001
LDL-C, mg/dL	120 ± 30	43 ± 6	62 ± 5	88 ± 8	110 ± 6	129 ± 6	159 ± 18	-
Glucose, mg/dL	100 ± 20	100 ± 12	100 ± 23	97 ± 18	100 ± 21	100 ± 19	102 ± 22	<0.001
Medications, *n* (%)								
Anti-hypertensive therapy	1669 (23.5)	10 (28.6)	49 (23.3)	312 (19.7)	443 (23.8)	460 (26.7)	395 (23.2)	<0.001
Anti-hyperglycemic therapy	311 (4.4)	3 (8.6)	13 (6.2)	70 (4.4)	80 (4.3)	70 (4.1)	75 (4.4)	0.617
Framingham risk score, %	8.1 ± 7.2	2.6 ± 2.0	2.7 ± 2.2	3.3 ± 2.9	7.9 ± 6.4	9.4 ± 7.0	12.3 ± 8.5	<0.001
Medical history, *n* (%)								
Hypertension	2924 (41.1)	13 (37.1)	77 (36.7)	543 (34.2)	767 (41.2)	769 (44.6)	755 (44.3)	<0.001
Dyslipidemia	2986 (41.9)	12 (34.3)	51 (24.3)	320 (20.2)	392 (21.1)	508 (29.5)	1703 (100.0)	<0.001
Diabetes mellitus	511 (7.2)	3 (8.6)	19 (9.1)	96 (6.1)	132 (7.1)	117 (6.8)	144 (8.5)	0.115
Smokers	2165 (30.5)	12 (34.3)	70 (33.3)	530 (33.4)	542 (29.2)	524 (30.5)	487 (28.7)	0.040
FMD, %	6.1 ± 3.2	6.2 ± 3.5	6.2 ± 3.2	6.5 ± 3.3	6.3 ± 3.3	5.9 ± 3.1	5.9 ± 3.0	<0.001

Data are presented as mean ± SD or median (interquartile range); LDL-C indicates low-density lipoprotein cholesterol; HDL-C, high-density lipoprotein cholesterol; FMD, flow-mediated vasodilation.

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
