# Peer review of "Low Levels of Low-Density Lipoprotein Cholesterol and Endothelial Function in Subjects without Lipid-Lowering Therapy"

_jcm, 2020, doi:10.3390/jcm9123796_

Round 1

Reviewer 1 Report

The study is a cross-sectional retrospective registry study. The aim is clear, the statistics mostly solid. Limitations are discussed. However, the interpretation should be improved. 

Statistical 

  • Why was Wilcoxon used in this analysis and not ANOVA? The Tukey's post hoc test requires similar varition in all groups, was this critera met?

Methodological 

  • It states that you have recorded the max velocity with doppler, but I cannot find the results, it should be added to the manuscript. Furthermore, there have been research showing added prognostic value to standardize the FMD response to the velocity (aka stimuli). Has any similar analysis been preformed with the velocity data? If not, why not? 
  • Also for the propensity score seems not to include data as physical activity or educational status. Factors which indeed are of great importance for cardiovascular risk but not necessarily connected to FMD, but should be included in a propensity score if available. Hence it is a very crude approximation. 

Results

  • Row 214, you state that LDL < 50mg/dL was independently associated with a lower quartile FMD, CL 1.00-4.56. I am under the understanding that CI containing the number 1 is not significant. You could possibly state borderline significance but should correspond to p value of =0.05. This should be changed and conclusion with it. 

Discussion

  • Row 240, You state that low LDL< 50mg/dL is independently associated with endothelial dysfunction. First, this is not entirely true as pointed out above. Second, your definition of endothelial dysfunction is not clear. Do you mean the lowest quartile in this material? I would argue that a lot of healthy people probably show the same level of FMD without condcidering it endothelial dysfunction. 
  • Propensity score matching was prone to bias and the results disappeared after multiple regression models when adjusting start, if there is a correlation it is very weak at best. Furthermore the overal FMD results was not different between the groups, even as the arms are quite large, hence it is unlikely that a clinicly real difference exist. 
  • Also the clinically relevance of this could be questioned. It hints that FMD is not linearly better with decreasing LDL-C but even if FMD is not improved lower LDL is clearly linked to  the atherosclerotic pathogenesis and decreasing it reduces the cardiovascular risk regardless if FMD change or not. 
  •  The conclusion mostly focuses on low LDL-C group. It should reflect the aim better with the above comments as addition. 

Author Response

Thank you for the thoughtful review of our manuscript by the editors and reviewers. The comments from the editor and the reviewers have been helpful in allowing us to revise our manuscript. We have attempted to address the questions raised by the reviewers as follows.

Manuscript ID: jcm-99003R1                                   Reviewer #1

We really appreciate the reviewer’s helpful and thoughtful comments. According to the reviewer’s advice, we believe that we have produced a more balanced and better account of our work.

  1. Statistical

Why was Wilcoxon used in this analysis and not ANOVA? The Tukey’s post hoc test requires similar variation in all groups, was this criteria met?

Response: In accordance with the reviewer’s suggestion, comparison of variables among two or more groups by differences in the LDL-C levels was performed using ANOVA rather than the Wilcoxon test or the Kruskal-Wallis test. After changing to the parametric test, the probability values were the same. We have therefore changed the sentence “Comparison of variables among the five groups by difference of the LDL-C levels was performed using the Wilcoxon test.” to “Comparison of variables among two or more groups by differences in the LDL-C levels was performed using one-way analysis of variance (ANOVA). Categorical values such as medications and medical histories were compared by means of the χ2 test.” in the Methods section (page 4, lines 145-147).

  1. Methodological

It states that you have recorded the max velocity with doppler, but I cannot find the results, it should be added to the manuscript. Furthermore, there have been research showing added prognostic value to standardize the FMD response to the velocity (aka stimuli). Has any similar analysis been performed with the velocity data? If not, why not?

Also for the propensity score seems not to include data as physical activity or educational status. Factors which indeed are of great importance for cardiovascular risk but not necessarily connected to FMD, but should be included in a propensity score if available. Hence it is a very crude approximation.

Response: We agree with the reviewer’s comment. Unfortunately, our database had limited information on max velocity. There was also no information on physical activity and educational status, so these could not be considered. Therefore, we have deleted the sentence “Pulsed Doppler flow was assessed at baseline and during peak hyperemic flow, which was confirmed to occur within 15 seconds after cuff deflation.” in the Methods section. In addition, we have added the sentence “Our database had limited information on physical activity and educational status. These confounders can affect the relationship between FMD and LDL-C levels.” in the Discussion section (page 14, lines 310-311).

  1. Results

Row 214, you state that LDL < 50mg/dL was independently associated with a lower quartile FMD, CL 1.00-4.56. I am the understanding that CI containing the number 1 is not significant. You could possibly state borderline significance but should correspond to p value of =0.05. This should be changed and conclusion with it.

Response: Following another reviewer’s suggestion that FMD should be analyzed as a continuous variable, we have deleted the results associated with odds ratio for the lowest quartile of FMD. The propensity score analysis was used to compare the <50 mg/dL group with all remaining participants (≥50 mg/dL). In this analysis, FMD was not significantly different between the LDL-C <50 mg/dL group and ≥50 mg/dL group in the propensity score-matched population (Table S3). These comments have been incorporated into the Results section (page 9, lines 223-226).

  1. Discussion

Row 240, you state that low LDL <50 mg/dL is independently associated with endothelial dysfunction. First, this is not entirely true as pointed out above. Second, your definition of endothelial dysfunction is not clear. Do you mean the lowest quartile in this material? I would argue that a lot of healthy people probably show the same level of FMD without considering it endothelial dysfunction.

Propensity score matching was prone to bias and the results disappeared after multiple regression models when adjusting start, if there is a correlation it is very weak at best.

Furthermore the overall FMD results was not different between the groups, even as the arms are quite large, hence it is unlikely that a clinically real difference exist.

Also the clinically relevance of this could be questioned. It hints that FMD is not linearly better with decreasing LDL-C but even if FMD is not improved lower LDL is clearly linked to  the atherosclerotic pathogenesis and decreasing it reduces the cardiovascular risk regardless if FMD change or not.

The conclusion mostly focuses on low LDL-C group. It should reflect the aim better with the above comments as addition.

Response: We have deleted the results associated with odds ratio for the lowest quartile of FMD. In the propensity score-matched population, there was no significant difference in FMD between low LDL-C groups and all remaining participants.

There are no accepted specific cut-offs that define endothelial dysfunction assessed by FMD. In this study, we excluded subjects with a history of cardiovascular disease and subjects taking lipid-lowering medicine, so it is possible that the lower quartile of FMD was high such as 4.0%. For this reason, we have deleted the expression that less than 4.0% is endothelial dysfunction in the Statistical Analysis section.

In addition, the sentences “In the present study, we demonstrated that low levels of LDL-C (<50 mg/dL) were independently associated with endothelial dysfunction after adjustment for other cardiovascular risk factors in subjects without a history of cardiovascular disease and not receiving lipid-lowering therapy. FMD values were highest in subjects with LDL-C of 70-99 mg/dL. Moreover, in the propensity score-matched population, FMD was significantly smaller in subjects with LDL-C of <70 mg/dL than in subjects with LDL-C of 70-99 mg/dL.” have been changed to “In the present study, we demonstrated that there was a significant inverse correlation between FMD and LDL-C levels (r=-0.07, p<0.001) in subjects without a history of cardiovascular disease and not receiving lipid-lowering therapy. Although there was not much difference in FMD among the groups categorized by LDL-C levels, FMD was highest in subjects with LDL-C of 70-99 mg/dL. FMD in the low LDL-C groups, including <70 mg/dL and <50 mg/dL groups, was not significantly different from that in the other groups. In the propensity score-matched population, there was no significant difference in FMD between the low LDL-C groups and all remaining participants.” in the Discussion section (page 12, lines 245-251).

Moreover, the sentences “Low levels of LDL-C (<50 mg/dL) in subjects not receiving lipid-lowering therapy and without previous cardiovascular disease were associated with endothelial dysfunction. Although the precise mechanisms of the association of low LDL-C and endothelial dysfunction is not known, it is unlikely that there would be more beneficial anti-atherosclerotic effects in subjects with LDL-C of 50 mg/dL who are not receiving lipid-lowering treatment and have no history of cardiovascular disease from the aspect of vascular function.” have been changed to “In the propensity score-matched population, there was no significant difference in FMD between low LDL-C groups and all remaining participants. Although the precise mechanisms of the association of low LDL-C and endothelial function is not known, a significant benefit was not observed in subjects with low LDL-C levels who were not receiving lipid-lowering treatment and had no history of cardiovascular disease from the aspect of endothelial function.” in the Conclusion section (page 14, lines 334-339).

Manuscript ID: jcm-99003R1                                  

The detailed review of this manuscript is appreciated, and we have attempted to answer the questions raised. Changes made are shown in red in the revised manuscript.

We hope that the revised manuscript is acceptable for publication in Journal of Clinical Medicine.

Sincerely,

Yukihito Higashi

Yukihito Higashi, MD, PhD, FAHA

Department of Cardiovascular Regeneration and Medicine,

Research Institute for Radiation Biology and Medicine,

Hiroshima University

1-2-3 Kasumi, Minami-ku, Hiroshima 734-8551, Japan

Phone: +81-82-257-5831

Fax: +81-82-257-5831 

Reviewer 2 Report

Takaeko et al. report data from two aggregated registry studies on the association of LDL-C with FMD, a marker of endothelial dysfunction. They conclude that low levels of LDL-C (<50 mg/dL) in individuals not on lipid-lowering drugs are associated with endothelial dysfunction.

The study's topic is relevant and tries to the question why low LDL-C levels are associated higher mortality in some association studies. 

However, there are major issues that need to be addressed by the authors, as in the current form, the conclusions are not fully supported by the results. This mainly is caused by an inaccurate description of the results, missing analyses and the statistical approach. 

A central issue is, that the authors base their conclusions on the (selected) analysis of odds for the lowest quartile of FMD (<4.0%) instead of the overall mean FMD of the respective groups. For FMD there are no accepted specific cut-offs that define endothelial dysfunction (such as <4.0%). Without accepted or established cut-offs, FMD should be analysed in studies as a continuous variable. Therefore also the multiple regression analyses should use FMD as continuous variable and not focus on the lowest quartile.

Page 4, Line 138-139: The authors state they compared the different variables between the LDL-C groups using the Wilcoxon test. As they compare 5 and not 2 groups, this is not the appropriate test for this question. For more than 2 groups, ANOVA (parametric data) or kruskal wallis test (non-parametric data) should be used. With such a large sample size, it is unclear why the authors chose a non-parametric over a parametric test. Is the data non-gaussian distributed? For FMD, they state that Tukey's post hoc test was used, albeit no information was given if ANOVA was performed at all.

Page 4, line 141 and 151: For the different statistical approaches, the authors defined different groups for the lower quartile of FMD (<4.0% for logistic regression but <4.2% for the propensity analysis).

Page 8, Line 197-198: The authors state FMD values were smaller in <70 mg/dL vs. 70-99 mg/dl and 100-119 mg/dL (referring to Table 1 and Figure 1). However, according to the figure 1, the difference is far from significant (p=0.776), so the accurate description in the text would be that there are no significant differences between the groups. Also in table 1, no post-hoc test for the difference between <70 vs 70-99 mg/dl are reported, so it does not help citing this table.

Page 8, Line 198-199: They then report a propensity score-matched analysis, however it is unclear which variables were exactly used for the propensity score. This should be stated in the description of Table 3. Despite the high sample size of the overall study, there are still important numerical differences in the propensity score matched analysis: notably, slightly higher age, higher use of anti-hypertensive lipid lowering therapy, higher rate of hypertension as well as diabetes and smokers. Although individuall, the differences were not significant, as a whole they may significantly confound the small difference in FMD that was found. Therefore, the reviewer suggests, that the propensity score analysis is used to compare the <70 mg/dL group with all remaining participants of the study (e.g. >=70mg/dL), which should improve the propensity matching.

Page 8, line 200-202: The authors show a local regression curve between FMD and LDL-C, and state that that "FMD decreased in relation to the decrease in LDL-C". However, no multiple regression analysis is shown, linking FMD with LDL-C (both as a continuous variables and not ordinal groups). Given the low sample size of participants with LDL <70 mg/dL, the confidence intervals in this region will likely be very high and will not show a significant result.

Page 8, line 202 ongoing: The authors then further decide to analyse the LDL-C groups <50mg/dL vs 50-69 mg/dL. Unfortunately, the supplementary tables were not accessible for the reviewer. But based on the text, the <50mg/dL group was very small and differed significantly in a wide range of important modifiers of endothelial function between the groups. They report that there were no significant differences in FMD between the groups. Instead of then reporting the results of another propensity score matched analysis of <50 vs 50-69 mg/dL, they then report a regression focusing on the lowest quartile of FMD, instead of reporting FMD as a continuous variable.

Table 2 and Table 4: The caption of both tables are inaccurate. They report the relationship of the odds ratio for the lowest quartile of FMD (yes vs no), but not the relationship between FMD and LDL-C (this would suggest continuous variable). As noted above, the analyses should be restricted on analysis FMD as a continuous variable.

Author Response

Thank you for the thoughtful review of our manuscript by the editors and reviewers. The comments from the editor and the reviewers have been helpful in allowing us to revise our manuscript. We have attempted to address the questions raised by the reviewers as follows.

Manuscript ID: jcm-99003R1                                    Reviewer #2

  1. A central issue is, that the authors base their conclusions on the (selected) analysis of odds for the lowest quartile of FMD (<4.0%) instead of the overall mean FMD of the respective groups. For FMD there are no accepted specific cut-offs that define endothelial dysfunction (such as <4.0%). Without accepted or established cut-offs, FMD should be analyzed in studies as a continuous variable. Therefore also the multiple regression analyses should use FMD as continuous variable and not focus on the lowest quartile.

Response: As you mentioned, there are no accepted specific cut-offs that define endothelial dysfunction. Therefore, FMD was reanalyzed as a continuous variable. We have deleted the results associated with odds ratio for the lowest quartile of FMD.

  1. Page 4, Line 138-139: The authors state they compared the different variables between the LDL-C groups using the Wilcoxon test. As they compare 5 and not 2 groups, this is not the appropriate test for this question. For more than 2 groups, ANOVA (parametric data) or kruskal wallis test (non-parametric data) should be used. With such a large sample size, it is unclear why the authors chose a non-parametric over a parametric test. Is the data non-gaussian distributed? For FMD, they state that Tukey's post hoc test was used, albeit no information was given if ANOVA was performed at all.

Response: In accordance with the reviewer’s appropriate suggestion, comparison of variables among two or more groups by differences in the LDL-C levels was performed using ANOVA rather than the Wilcoxon test or the Kruskal-Wallis test. After changing to the parametric test, the probability values were almost the same. We therefore have changed the sentence “Comparison of variables among the five groups by difference of the LDL-C levels was performed using the Wilcoxon test.” to “Comparison of variables among two or more groups by differences in the LDL-C levels was performed using one-way analysis of variance (ANOVA). Categorical values such as medications and medical histories were compared by means of the χ2 test.” in the Methods section (page 4, lines 145-147)

  1. Page 4, line 141 and 151: For the different statistical approaches, the authors defined different groups for the lower quartile of FMD (<4.0% for logistic regression but <4.2% for the propensity analysis).

Response: In accordance with the reviewer’s appropriate suggestion, we have deleted the results associated with odds ratio for the lowest quartile of FMD. In the propensity score analysis, FMD of <4.0%, which was lowest quartile of FMD in all subjects, was used.

  1. Page 8, Line 197-198: The authors state FMD values were smaller in <70 mg/dL vs. 70-99 mg/dl and 100-119 mg/dL (referring to Table 1 and Figure 1). However, according to the figure 1, the difference is far from significant (p=0.776), so the accurate description in the text would be that there are no significant differences between the groups. Also in table 1, no post-hoc test for the difference between <70 vs 70-99 mg/dl are reported, so it does not help citing this table.

Response: In accordance with the reviewer’s appropriate suggestion, the sentence “The FMD values in the LDL-C <70 mg/dL group was smaller than the values in the 70-99 mg/dL group and 100-110 mg/dL group (Table1 and Figure1).” has been changed to “The FMD values in the LDL-C <70 mg/dL group were not significantly different from those in the other groups (Table 1 and Figure 2).” in the Results section (page 10, line 202-203).

  1. Page 8, Line 198-199: They then report a propensity score-matched analysis, however it is unclear which variables were exactly used for the propensity score. This should be stated in the description of Table 3. Despite the high sample size of the overall study, there are still important numerical differences in the propensity score matched analysis: notably, slightly higher age, higher use of anti-hypertensive lipid lowering therapy, higher rate of hypertension as well as diabetes and smokers. Although individuall, the differences were not significant, as a whole they may significantly confound the small difference in FMD that was found. Therefore, the reviewer suggests, that the propensity score analysis is used to compare the <70 mg/dL group with all remaining participants of the study (e.g. >=70mg/dL), which should improve the propensity matching.

Response: In accordance with the reviewer’s appropriate suggestion, the propensity score analysis was used to compare the <70 mg/dL group with all remaining participants (≥70 mg/dL). In this analysis, FMD was not significantly different between the LDL-C <70 mg/dL group and ≥70 mg/dL group in the propensity score-matched population (p=0.386). We have added the variables used for propensity score to the description in Table 3.

  1. Page 8, line 200-202: The authors show a local regression curve between FMD and LDL-C, and state that that "FMD decreased in relation to the decrease in LDL-C". However, no multiple regression analysis is shown, linking FMD with LDL-C (both as a continuous variables and not ordinal groups). Given the low sample size of participants with LDL <70 mg/dL, the confidence intervals in this region will likely be very high and will not show a significant result.

Response: As the reviewer pointed out, there was a small sample size of participants with LDL-C <70 mg/dL. We have deleted the estimated Lowess smoothed curve and added results of univariate analysis of the relationships between FMD and variables including LDL-C (Figure 1). In addition, we have added results of multiple regression analysis (Table 2).

  1. Page 8, line 202 ongoing: The authors then further decide to analyse the LDL-C groups <50mg/dL vs 50-69 mg/dL. Unfortunately, the supplementary tables were not accessible for the reviewer. But based on the text, the <50mg/dL group was very small and differed significantly in a wide range of important modifiers of endothelial function between the groups. They report that there were no significant differences in FMD between the groups. Instead of then reporting the results of another propensity score matched analysis of <50 vs 50-69 mg/dL, they then report a regression focusing on the lowest quartile of FMD, instead of reporting FMD as a continuous variable.

Response: In accordance with the reviewer’s appropriate suggestion, we have deleted the results associated with odds ratio for the lowest quartile of FMD. The propensity score analysis was used to compare the <50 mg/dL group with all remaining participants (≥50 mg/dL). In this analysis, FMD was not significantly different between the LDL-C <50 mg/dL group and ≥50 mg/dL group in the propensity score-matched population (p=0.570) (Table S3) in the Results section (page 10, lines 223-226).

  1. Table 2 and Table 4: The caption of both tables are inaccurate. They report the relationship of the odds ratio for the lowest quartile of FMD (yes vs no), but not the relationship between FMD and LDL-C (this would suggest continuous variable). As noted above, the analyses should be restricted on analysis FMD as a continuous variable.

Response: In accordance with the reviewer’s appropriate suggestion, we have deleted the results associated with odds ratio for the lowest quartile of FMD. In the propensity score-matched population, there was no significant difference in FMD values between the low LDL-C group and the remaining participants. Therefore, the sentences “In the present study, we demonstrated that low levels of LDL-C (<50 mg/dL) were independently associated with endothelial dysfunction after adjustment for other cardiovascular risk factors in subjects without a history of cardiovascular disease and not receiving lipid-lowering therapy. FMD values were highest in subjects with LDL-C of 70-99 mg/dL. Moreover, in the propensity score-matched population, FMD was significantly smaller in subjects with LDL-C of <70 mg/dL than in subjects with LDL-C of 70-99 mg/dL.” have been changed to “In the present study, we demonstrated that there was a significant inverse correlation between FMD and LDL-C levels (r=-0.07, p<0.001) in subjects without a history of cardiovascular disease and not receiving lipid-lowering therapy. Although there was not much difference in FMD among the groups categorized by LDL-C levels, FMD was highest in subjects with LDL-C of 70-99 mg/dL. FMD in the low LDL-C groups, including <70 mg/dL and <50 mg/dL groups, was not significantly different from that in the other groups. In the propensity score-matched population, there was no significant difference in FMD between the low LDL-C groups and all remaining participants.” in the Discussion section (page 12, lines 245-251). In addition, the sentences “Low levels of LDL-C (<50 mg/dL) in subjects not receiving lipid-lowering therapy and without previous cardiovascular disease were associated with endothelial dysfunction. Although the precise mechanisms of the association of low LDL-C and endothelial dysfunction is not known, it is unlikely that there would be more beneficial anti-atherosclerotic effects in subjects with LDL-C of 50 mg/dL who are not receiving lipid-lowering treatment and have no history of cardiovascular disease from the aspect of vascular function.” have been changed to “In the propensity score-matched population, there was no significant difference in FMD between low LDL-C groups and all remaining participants. Although the precise mechanisms of the association of low LDL-C and endothelial function is not known, a significant benefit was not observed in subjects with low LDL-C levels who were not receiving lipid-lowering treatment and had no history of cardiovascular disease from the aspect of endothelial function.” in the Conclusion section (page 14, lines 334-339).

Manuscript ID: jcm-99003R1                                  

The detailed review of this manuscript is appreciated, and we have attempted to answer the questions raised. Changes made are shown in red in the revised manuscript.

We hope that the revised manuscript is acceptable for publication in Journal of Clinical Medicine.

Sincerely,

Yukihito Higashi

Yukihito Higashi, MD, PhD, FAHA

Department of Cardiovascular Regeneration and Medicine,

Research Institute for Radiation Biology and Medicine,

Hiroshima University

1-2-3 Kasumi, Minami-ku, Hiroshima 734-8551, Japan

Phone: +81-82-257-5831

Fax: +81-82-257-5831 

Reviewer 3 Report

The present study by Takaeko et al. titled: “Low Levels of Low-density Lipoprotein Cholesterol and Endothelial Function in Subjects without Lipid-lowering Therapy”, is a cross-sectional study from the FMD-Japan registry and from Hiroshima University Vascular Function registry, investigating the association between low-density lipoprotein cholesterol and endothelial function, as measured by flow mediated dilation.

The authors conclude that low levels of LDL-C (<50 mg/dL) were independently associated with endothelial dysfunction after adjustment for other cardiovascular risk factors in subjects without a history of cardiovascular disease and not receiving lipid-lowering therapy.

This article may be of potential interest to the readers of Journal of Clinical Medicine; however, the authors should consider the following pertinent points.

  1. Flow mediated dilation (FMD) has been widely used to assess endothelial function in vivo. Some considerations regarding this method: In general, greater accuracy in measurement is achieved with simultaneous longitudinal and cross-section measurement of FMD. Another issue with FMD is reproducibility. The Authors should also add Bland-Altman plots (with standard errors), along with the high correlations proved in the text.
  2. More details are needed for the conditions under which FMD was measured. There have been extensive reports in the literature regarding setting factors that may affect measurements (eg. room temperature). In addition, FMD can also be influenced by caffeine consumption or ovulation. More detail is needed in terms of how that was addressed in the study and what patients were included for the measurements.
  3. Since this is one of the main conclusions of the study, the authors should add a very detailed table in the main text, with the demographic characteristics for each sub-group of the “extremely low LDL-C” group (<50, 50-69).
  4. The authors should address the limitations of propensity score group matching in the study limitations section, as unrecognized confounders still remain even after propensity score matching.

Author Response

Thank you for the thoughtful review of our manuscript by the editors and reviewers. The comments from the editor and the reviewers have been helpful in allowing us to revise our manuscript. We have attempted to address the questions raised by the reviewers as follows.

Manuscript ID: jcm-99003R1                                    Reviewer #3

  1. Flow mediated dilation (FMD) has been widely used to assess endothelial function in vivo. Some considerations regarding this method: In general, greater accuracy in measurement is achieved with simultaneous longitudinal and cross-section measurement of FMD. Another issue with FMD is reproducibility. The Authors should also add Bland-Altman plots (with standard errors), along with the high correlations proved in the text.

Response: While the longitudinal image of the artery was recorded, the short-axis image was also observed at the same time. However, we did not achieved cross-section measurement of FMD. We have added the sentence “At the same time as recording the longitudinal image of the artery, we also observed the short-axis image using a high resolution ultrasonography system.” into the Materials and Methods section (page 4, lines127-129). Moreover, we added the sentence “We confirmed that Bland-Altman analysis showed no systemic bias in the FMD measurement variability between the institutions and the core laboratory.” into the Materials and Methods section (page 4, lines134-136). The following new reference [33] has been added.

[33] Tomiyama, H., et al. Reliability of measurement of endothelial function across multiple institutions and establishment of reference values in Japanese. Atherosclerosis 2015, 242, 433-442, doi:10.1016/j.atherosclerosis.2015.08.001.

  1. More details are needed for the conditions under which FMD was measured. There have been extensive reports in the literature regarding setting factors that may affect measurements (eg. room temperature). In addition, FMD can also be influenced by caffeine consumption or ovulation. More detail is needed in terms of how that was addressed in the study and what patients were included for the measurements.

Response: In accordance with the reviewer’s appropriate suggestion, we have added the sentence “Subjects fasted overnight for at least 12 hours and abstained from caffeine, alcohol, smoking, and antioxidant vitamins on the day of the FMD examination. The subjects were kept in the supine position in a quiet, dark, air-conditioned room (constant temperature of 22-25°C) through the study.” into the Materials and Methods section (page 3, lines 111-113).

 In addition, we have added the sentences “The FMD-J study was a prospective multicenter study conducted at 22 university hospital and affiliated clinics in Japan to examine the usefulness of flow-mediated vasodilation in risk stratification for cardiovascular disease in Japanese subjects. The rationale and design of the FMD-J study have been described previously.” into the Materials and Methods section (page 3, lines 86-90). The following new reference [29] has been added.

[29] Tomiyama, H., et al. A multicenter study design to assess the clinical usefulness of semi-automatic measurement of flow-mediated vasodilatation of the brachial artery. International heart journal 2012, 53, 170-175, doi:10.1536/ihj.53.170.

  1. Since this is one of the main conclusions of the study, the authors should add a very detailed table in the main text, with the demographic characteristics for each sub-group of the “extremely low LDL-C” group (<50, 50-69).

Response: In accordance with the reviewer’s appropriate suggestion, we have moved baseline characteristics of extremely low LDL-C group described in the Supplemental material to the main text (Table 4).

  1. The authors should address the limitations of propensity score group matching in the study limitations section, as unrecognized confounders still remain even after propensity score matching.

Response: In accordance with the reviewer’s appropriate suggestion, we have added the sentence “Confounding factors may remain even after the propensity score match.” into the Discussion section (page 14, line 309).

  1.  

Manuscript ID: jcm-99003R1                                  

The detailed review of this manuscript is appreciated, and we have attempted to answer the questions raised. Changes made are shown in red in the revised manuscript.

We hope that the revised manuscript is acceptable for publication in Journal of Clinical Medicine.

Sincerely,

Yukihito Higashi

Yukihito Higashi, MD, PhD, FAHA

Department of Cardiovascular Regeneration and Medicine,

Research Institute for Radiation Biology and Medicine,

Hiroshima University

1-2-3 Kasumi, Minami-ku, Hiroshima 734-8551, Japan

Phone: +81-82-257-5831

Fax: +81-82-257-5831 
